# CONFORMAL PREDICTION IS ROBUST TO LABEL NOISE

## ABSTRACT

We study the robustness of conformal prediction—a powerful tool for uncertainty quantification—to label noise. Our analysis tackles both regression and classification problems, characterizing when and how it is possible to construct uncertainty sets that correctly cover the unobserved noiseless ground truth labels. With both theory and experiments, we argue that conformal prediction with noisy labels conservatively covers the clean ground truth labels except in adversarial cases. This leads us to believe that correcting for label noise is unnecessary except for pathological data distributions or noise sources. In such cases, we can also correct for noise of bounded size in the conformal prediction algorithm in order to ensure correct coverage of the ground truth labels without score or data regularity.

## 1 INTRODUCTION

In most supervised classification and regression tasks, one would assume the provided labels reflect the ground truth. In reality, this assumption is often violated; see (Cheng et al., 2022; Xu et al., 2019; Yuan et al., 2018; Lee & Barber, 2022; Cauchois et al., 2022). For example, doctors labeling the same medical image may have different subjective opinions about the diagnosis, leading to variability in the ground truth label itself. In other settings, such variability may arise due to sensor noise, data entry mistakes, the subjectivity of a human annotator, or many other sources. In other words, the labels we use to train machine learning (ML) models may often be noisy in the sense that these are not necessarily the ground truth. Quantifying the prediction uncertainty is crucial in high-stakes applications in general, and especially so in settings where the training data is inexact. We aim to investigate uncertainty quantification in this challenging noisy setting via conformal prediction, a framework that uses hold-out calibration data to construct prediction sets that are guaranteed to contain the ground truth labels; see (Vovk et al., 2005; Angelopoulos & Bates, 2021). In short, this paper shows that conformal prediction typically yields confidence sets with conservative coverage when the hold-out calibration data has noisy labels.

We adopt a variation of the standard conformal prediction setup. Consider a *calibration data set* of i.i.d. observations $\{(X_i, Y_i)\}_{i=1}^n$ sampled from an arbitrary unknown distribution $P_{XY}$. Here, $X_i \in \mathbb{R}^p$ is the feature vector that contains $p$ features for the $i$th sample, and $Y_i$ denotes its response, which can be discrete for classification tasks or continuous for regression tasks. Given the calibration dataset, an i.i.d. test data point $(X_{\text{test}}, Y_{\text{test}})$, and a pre-trained model $\hat{f}$, conformal prediction constructs a set $\widehat{\mathcal{C}}(X_{\text{test}})$ that contains the unknown test response, $Y_{\text{test}}$, with high probability, e.g., 90%. That is, for a user-specified level $\alpha \in (0, 1)$,

$$\mathbb{P}\left(Y_{\text{test}} \in \widehat{\mathcal{C}}(X_{\text{test}})\right) \geq 1 - \alpha. \tag{1}$$

This property is called *marginal coverage*, where the probability is defined over the calibration and test data.

In the setting of label noise, we only observe the corrupted labels $\tilde{Y}_i = g(Y_i)$ for some *corruption function* $g : \mathcal{Y} \times [0, 1] \to \mathcal{Y}$, so the i.i.d. assumption and marginal coverage guarantee are invalidated. The corruption is random; we will always take the second argument of $g$ to be a random seed $U$ uniformly distributed on $[0, 1]$. To ease notation, we leave the second argument implicit henceforth. Nonetheless, using the noisy calibration data, we seek to form a prediction set $\widehat{\mathcal{C}}_{\text{noisy}}(X_{\text{test}})$ that covers the clean, uncorrupted test label, $Y_{\text{test}}$. More precisely, our goal is to delineate when it is

possible to provide guarantees of the form

$$\mathbb{P}\left(Y_{\text{test}} \in \widehat{\mathcal{C}}_{\text{noisy}}(X_{\text{test}})\right) \geq 1 - \alpha, \tag{2}$$

where the probability is taken jointly over the calibration data, test data, and corruption function (this will be the case for the remainder of the paper). Our theoretical vignettes and experiments suggest that in realistic situations, (2) is usually satisfied. That is, even with access only to noisy labels, conformal prediction yields confidence sets that have conservative coverage on clean labels. There are a few failure cases involving adversarial noise that we discuss, but in general we argue that a user should feel safe deploying conformal prediction even with noisy labels.

### MOTIVATIONAL EXAMPLE

As a real-world example of label noise, we conduct an image classification experiment where we only observe one annotater's label but seek to cover the majority vote of many annotators. For this purpose, we use the CIFAR-10H data set, first introduced by Peterson et al. (2019); Battleday et al. (2020); Singh et al. (2020), which contains 10,000 images labeled by approximately 50 annotators. We calibrate using only a single annotator and seek to cover the majority vote of the 50. The single annotator differs from the ground truth labels in approximately 5% of the images.

Using the noisy calibration set (i.e., a calibration set containing these noisy labels), we applied vanilla conformal prediction as if the data were i.i.d, and studied the performance of the resulting prediction sets. Details regarding the training procedure can be found in section 4.2. The fraction of majority vote labels covered is demonstrated in Figure 1. As we can see, when using the clean calibration set the marginal coverage is 90%, as expected. When using the noisy calibration set, the coverage increases to approximately 93%. Figure 1 also demonstrates the prediction sets that are larger when calibrating with noisy labels. This experiment demonstrates the main intuition behind our paper: adding noise will usually increase the variability in the labels, leading to larger prediction sets that retain the coverage property.

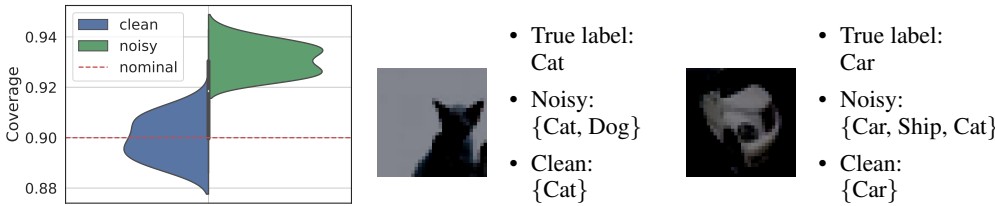

Figure 1: **Effect of label noise on CIFAR-10.** Left: distribution of average coverage on a clean test set over 30 independent experiments evaluated on CIFAR-10H test data with target coverage $1 - \alpha = 90\%$, using noisy and clean labels for calibration. We use a pre-trained resnet 18 model, which has Top-1 accuracy of 93% and 90% on the clean and noisy test set, respectively. The gray bar represents the interquartile range. Center and right: prediction sets achieved using noisy and clean labels for calibration.

## 2 THEORETICAL ANALYSIS

In this section we show mathematically that under stylized settings and some regularity conditions, the marginal coverage guarantee (1) of conformal prediction persists even when the labels used for calibration are noisy; i.e., (2) holds. In Sections 3 and 4 we support this argument with realistic experiments. Towards that end, we now give more details on the conformal prediction algorithm.

As explained in the introduction, conformal prediction uses a held-out calibration data set and a pre-trained model to construct the prediction set on a new data point. More formally, we use the model $\hat{f}$ to construct a score function, $s : \mathcal{X} \times \mathcal{Y} \to \mathbb{R}$, which is engineered to be large when the model is uncertain and small otherwise.

Abbreviate the scores on each calibration data point as $s_i = s(X_i, Y_i)$ for each $i = 1, ..., n$. Conformal prediction tells us that we can achieve a marginal coverage guarantee by picking

$\hat{q}_{\text{clean}} = s_{(\lceil (n+1)(1-\alpha) \rceil)}$ as the $\lceil (n+1)(1-\alpha) \rceil$-smallest of the calibration scores and constructing the prediction sets as

$$\widehat{\mathcal{C}}(X_{\text{test}}) = \{y \in \mathcal{Y} : s(X_{\text{test}}, y) \leq \hat{q}_{\text{clean}}\}.$$

We will introduce different score functions for both classification and regression as needed in the following subsections.

In this paper, we do not allow ourselves access to the calibration labels, only their noisy versions, $\tilde{Y}_1, \ldots, \tilde{Y}_n$, so we cannot calculate $\hat{q}_{\text{clean}}$. Instead, we can calculate the noisy quantile $\hat{q}_{\text{noisy}}$ as the $\lceil (n+1)(1-\alpha) \rceil$-smallest of the *noisy* score functions, $\tilde{s}_i = s(X_i, \tilde{Y}_i)$. The main formal question of our work is whether the resulting prediction set, $\widehat{\mathcal{C}}_{\text{noisy}}(X_{\text{test}}) = \{y : s(X_{\text{test}}, y) \leq \hat{q}_{\text{noisy}}\}$, covers the clean label as in (2). We state this general recipe algorithmically for future reference:

**Recipe 1** (Conformal prediction with noisy labels)**.**

1. *Consider i.i.d. data points* $(X_1, Y_1), \ldots, (X_n, Y_n), (X_{\text{test}}, Y_{\text{test}})$, *a corruption model* $g : \mathcal{Y} \to \mathcal{Y}$, *and a score function* $s : \mathcal{X} \times \mathcal{Y} \to \mathbb{R}$.

2. *Compute the conformal quantile with the corrupted labels,*

$$\hat{q}_{\text{noisy}} = \text{Quantile}\left(\frac{(n+1)(1-\alpha)}{n}, \{s(X_i, \tilde{Y}_i)\}_{i=1}^n\right),$$

*where* $\tilde{Y}_i = g(Y_i)$.

3. *Construct the prediction set using the noisy conformal quantile,*

$$\widehat{\mathcal{C}}_{\text{noisy}} = \{y : s(X_{\text{test}}, y) \leq \hat{q}_{\text{noisy}}\}.$$

This recipe will produce valid prediction sets whenever the noisy score distribution stochastically dominates the clean score distribution. The intuition is that the noise distribution 'spreads out' the distribution of the score function such that $\hat{q}_{\text{noisy}}$ is (stochastically) larger than $\hat{q}_{\text{clean}}$.

**Theorem 1.** *Assume that* $\mathbb{P}(\tilde{s}_{\text{test}} \leq t) \leq \mathbb{P}(s_{\text{test}} \leq t)$. *Then,*

$$\mathbb{P}\left(Y_{\text{test}} \in \widehat{\mathcal{C}}_{\text{noisy}}(X_{\text{test}})\right) \geq 1 - \alpha.$$

*Furthermore, for any* $u$ *satisfying* $\mathbb{P}(\tilde{s}_{\text{test}} \leq t) + u \geq \mathbb{P}(s_{\text{test}} \leq t)$, *then*

$$\mathbb{P}\left(Y_{\text{test}} \in \widehat{\mathcal{C}}_{\text{noisy}}(X_{\text{test}})\right) \leq 1 - \alpha + \frac{1}{n+1} + u.$$

In the following subsections, we present examples under which this schochastic dominance holds, and conformal prediction with noisy labels succeeds in covering the true, noiseless label. The purpose of these examples is to illustrate simple and intuitive statistical setups where Theorem 1 holds. Under the hood, all proofs from Sections 2.1– 2.3 are applications of Theorem 1. Though the noise can be adversarially designed to violate these assumptions and cause miscoverage (as in the impossibility result Proposition 1), the evidence presented here suggests that in the majority of practical settings, conformal prediction can be applied without modification. All proofs are in Appendix A1.

## 2.1 RANDOM CORRUPTIONS IN CLASSIFICATION

First we will examine a classification problem similar to our motivating example in Section 1, in which the noisy label is randomly flipped $\epsilon$ fraction of the time. This noise model is well-studied in the literature; see, for example, (Aslam & Decatur, 1996; Angluin & Laird, 1988; Ma et al., 2018; Jenni & Favaro, 2018; Jindal et al., 2016; Yuan et al., 2018). Formally, in the setting of $K$-class classification, we define the corruption model as follows:

$$g^{\text{flip}}(y) = \begin{cases} y & \text{w.p } 1 - \epsilon \\ Y' & \text{else,} \end{cases}$$

where $Y'$ is uniformly drawn from the set $\{1, ..., K\}$. Also consider the *adaptive prediction sets* (APS) scores, first introduced by Romano et al. (2020),

$$s^{\text{APS}}(x, y) = \sum_{y' \in \mathcal{Y}} \hat{\pi}_{y'}(x) \, \mathbb{I}\{\hat{\pi}_{y'}(x) > \hat{\pi}_y(x)\} + \hat{\pi}_y(x) \cdot U',$$

where $\mathbb{I}$ is the indicator function, $\hat{\pi}_y(x)$ is the estimated conditional probability $\mathbb{P}(Y = y \mid X = x)$ and $U' \sim \text{Unif}(0, 1)$. To make this non-random, a variant of the above where $U' = 1$ is often used.

The next proposition shows that for any classifier that ranks classes in the same order as $\mathbb{P}(\tilde{Y} \mid X)$ the APS score leads to valid coverage even when calibrated on noisy data.

**Example 1.** *Let $\widehat{\mathcal{C}}_{\text{noisy}}^{\text{APS}}(X_{\text{test}})$ be constructed as in Recipe 1 with the corruption function $g^{\text{flip}}$ and the score function $s^{\text{APS}}$ (deterministic version) with any classifier that ranks the classes in the same order as the oracle classifier $\hat{\pi}_y(x) = \mathbb{P}(\tilde{Y} = y \mid X = x)$. Then*

$$1 - \alpha \leq \mathbb{P}\Big(Y_{\text{test}} \in \widehat{\mathcal{C}}_{\text{noisy}}^{\text{APS}}(X_{\text{test}})\Big) \leq 1 - \alpha + \frac{1}{n+1} + \epsilon \frac{K-1}{K}.$$

The APS score is one of two popular conformal methods for classification. The other score from Vovk et al. (2005); Lei et al. (2013), is referred to as *homogeneous prediction sets* (HPS) score, $s^{\text{HPS}}(x, y) = 1 - \hat{\pi}_y(x)$, for some classifier $\hat{\pi}_y(x) \in [0, 1]$. The next proposition shows that with access to an oracle classifier for the noisy label distribution $\hat{\pi}_y(x) = \mathbb{P}(\tilde{Y} = y \mid X = x)$, conformal prediction covers the noiseless test label.

**Example 2.** *Let $\widehat{\mathcal{C}}_{\text{noisy}}^{\text{HPS}}(X_{\text{test}})$ be constructed as in Recipe 1 with the corruption function $g^{\text{flip}}$ where $\epsilon \leq (1 - \frac{1}{K})^2$, and the score function $s^{\text{HPS}}$ with the oracle classifier $\hat{\pi}_y(x) = \mathbb{P}(\tilde{Y} = y \mid X = x)$. Then*

$$1 - \alpha \leq \mathbb{P}\Big(Y_{\text{test}} \in \widehat{\mathcal{C}}_{\text{noisy}}^{\text{HPS}}(X_{\text{test}})\Big) \leq 1 - \alpha + \frac{1}{n+1} + \frac{\epsilon^2}{1 - \epsilon}(K - 1).$$

It should be noted that the above theorem only requires knowledge of the *noisy* conditional distribution $\mathbb{P}(\tilde{Y} \mid X)$, so a model trained on a large amount of noisy data should approximately have the desired coverage as well.

## 2.2 Conformalized quantile regression with symmetric noise

Next we will analyze a regression task where the labels are continuous-valued and the corruption function is

$$g^{\text{sym}}(y) = y + Z$$

for some symmetric, unimodal, independent noise sample $Z$. We also assume that $Y \mid X$ is symmetric unimodal. We analyze the most common regression strategy, conformalized quantile regression (CQR) as presented by Romano et al. (2019). Let the score function be

$$s^{\text{CQR}}(x, y) := \max\{\hat{q}_{\alpha_{lo}}(x) - y, y - \hat{q}_{\alpha_{hi}}(x), 0\}. \tag{3}$$

where $\hat{q}_{\alpha_{lo}}$ and $\hat{q}_{\alpha_{hi}}$ are estimates of the $\alpha/2$ and $1 - \alpha/2$ conditional quantiles respectively.

We will analyze the setting where $\hat{q}_{\alpha_{lo}}$ and $\hat{q}_{\alpha_{hi}}$ fall on the correct side of the true median of the noisy distribution, an extremely weak assumption about the fitted model. In this case, conformalized quantile regression achieves valid coverage even with noisy labels, as stated next.

**Example 3.** *Let $\widehat{\mathcal{C}}_{\text{noisy}}^{\text{CQR}}(X_{\text{test}})$ be constructed as in Recipe 1 with the corruption function $g^{\text{sym}}$ and the score function $s^{\text{CQR}}$ where the estimated quantiles satisfy $\hat{q}_{\alpha_{lo}}(x) \leq q_{1/2}(x) \leq \hat{q}_{\alpha_{hi}}(x)$, where $q_{1/2}(x)$ is the true median of $\tilde{Y} \mid X = x$. Then*

$$\mathbb{P}\Big(Y_{\text{test}} \in \widehat{\mathcal{C}}_{\text{noisy}}^{\text{CQR}}(X_{\text{test}})\Big) \geq 1 - \alpha.$$

## 2.3 Conformalized uncertainty scalars

We next analyze the case of a score function using a single uncertainty scalar, $\hat{u}(x) > 0$. In this setting, conformal prediction will work whenever the noisy label distribution has the same mean as, but heavier tails than, the ground truth label distribution.

We set the corruption function to be

$$g^{\text{vi}}(y) = y + Z,$$

where $Z$ satisfies

$$\mathbb{P}\Big(\big|Y - \mathbb{E}\big[Y \mid X = x\big]\big| > t \,\Big|\, X = x\Big) \leq \mathbb{P}\Big(\big|Y + Z - \mathbb{E}\big[Y \mid X = x\big]\big| > t \,\Big|\, X = x\Big).$$

This condition is satisfied in many statistical setups, including the case from the previous section, where $Y$ and $Z$ are symmetric and $Y$ is unimodal.

Also take the score function to be the normalized magnitude of the residual,

$$s^{\mathrm{RM}}(x, y) = \big|\hat{f}(x) - y\big| / \hat{u}(x).$$

The following result states that conformal prediction is robust to these conditions.

**Example 4.** *Let $\widehat{\mathcal{C}}^{\mathrm{RM}}_{\mathrm{noisy}}$ be constructed as in Recipe 1 with the score function $s^{\mathrm{RM}}$ using the oracle model $\hat{f}(x) = \mathbb{E}[\tilde{Y}_i \mid X_i = x]$ and the corruption function $g^{\mathrm{vi}}$. Then*

$$\mathbb{P}\Big(Y_{\mathrm{test}} \in \widehat{\mathcal{C}}^{\mathrm{RM}}_{\mathrm{noisy}}(X_{\mathrm{test}})\Big) \geq 1 - \alpha.$$

This example summarizes the practical intuition that if $Y$ has lighter tails than $Y + Z$, then conformal prediction will be conservative (see Corollary 3 in Appendix A1 for a statement in terms of tail bounds).

### 2.4 DISCLAIMER: DISTRIBUTION-FREE RESULTS

Though the coverage guarantee holds in many realistic cases, we have also given examples where conformal prediction fails to cover. Indeed, in the general case, conformal prediction will fail to cover, and must be adjusted to account for the size of the noise. The following proposition makes it clear that for any nontrivial noise distribution, there exists a score function that breaks naïve conformal.

**Proposition 1** (Coverage is impossible in the general case.)**.** *Take any $\tilde{Y} \overset{d}{\neq} Y$. Then there exists a score function $s$ that yields $\mathbb{P}\Big(Y_{\mathrm{test}} \in \widehat{\mathcal{C}}_{\mathrm{noisy}}(X_{\mathrm{test}})\Big) < \mathbb{P}\Big(Y_{\mathrm{test}} \in \widehat{\mathcal{C}}(X_{\mathrm{test}})\Big).$*

The above proposition says that for any noise distribution, there exists an adversarially chosen score function that will disrupt coverage. Furthermore, as we discuss in Appendix A1, with noise of a sufficient magnitude, it is possible to get arbitrarily bad violations of coverage.

Next, we discuss how to adjust the threshold of conformal prediction to account for noise of a known size, as measured by total variation (TV) distance from the clean label.

**Corollary 1** (Corollary of Barber et al. (2022))**.** *Let $\tilde{Y}$ be any random variable satisfying $D_{\mathrm{TV}}(Y, \tilde{Y}) \leq \epsilon$. Take $\alpha' = \alpha + \frac{n}{n+1}\epsilon$. Letting $\widehat{\mathcal{C}}_{\mathrm{noisy}}(X_{\mathrm{test}})$ be the output of Recipe 1 with any score function at level $\alpha'$ yields*

$$\mathbb{P}\big(Y_{\mathrm{test}} \in \widehat{\mathcal{C}}_{\mathrm{noisy}}(X_{\mathrm{test}})\big) \geq 1 - \alpha.$$

We discuss this strategy more in Appendix A1—the algorithm implied by Corollary 1 may not be particularly useful, as the TV distance is a badly behaved quantity that is also difficult to estimate.

As a final note, if the noise is bounded in TV norm, then the coverage is also not too conservative.

**Corollary 2** (Corollary of Barber et al. (2022) Theorem 3)**.** *Let $\tilde{Y}$ be any random variable satisfying $D_{\mathrm{TV}}(Y, \tilde{Y}) \leq \xi$. Letting $\widehat{\mathcal{C}}_{\mathrm{noisy}}(X_{\mathrm{test}})$ be the output of Recipe 1 with any score function at level $\alpha$ yields*

$$\mathbb{P}\big(Y_{\mathrm{test}} \in \widehat{\mathcal{C}}_{\mathrm{noisy}}(X_{\mathrm{test}})\big) \leq 1 - \alpha + \frac{1}{n+1} + \frac{n}{n+1}\xi.$$

## 3 SYNTHETIC EXPERIMENTS

### 3.1 CLASSIFICATION

In this section, we focus on multi-class classification problems, where we study the validity of conformal prediction using different types of label noise distributions, described below.

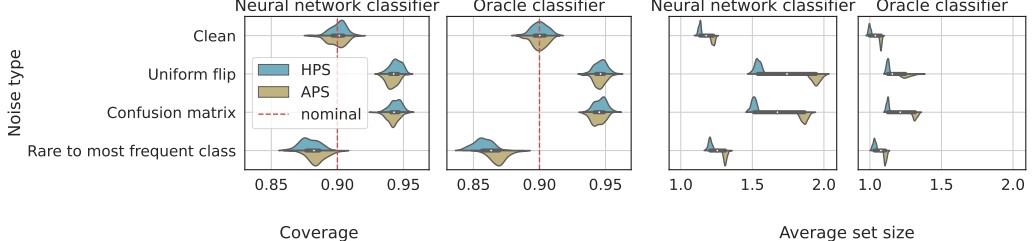

Figure 2: **Effect of label noise on synthetic multi-class classification data**. Performance of conformal prediction sets with target coverage $1 - \alpha = 90\%$, using a noisy training set and a noisy calibration set. Left: Marginal coverage; Right: Average size of predicted sets. The results are evaluated over 100 independent experiments and the gray bar represents the interquartile range.

**Class-independent noise.** This noise model, which we call `uniform flip`, randomly flips the ground truth label with probability $\epsilon$, defined formally by the class corruption function $g^{\text{flip+}}$ in (A6). We analyzed this setting in depth in Section 2.1, and proved that the coverage achieved by an oracle classifier is guaranteed to increase.

**Class-dependent noise.** In contrast to `uniform flip` noise, here we consider a more challenging setup in which the probability of a label to be flipped depends on the ground truth class label $Y$. Such a noise label is often called Noisy at Random (NAR) in the literature, where certain classes are more likely to be mislabeled or confused with similar ones. Let $T$ be a row stochastic transition matrix of size $K \times K$ such that $T_{i,j}$ is the probability of a point with label $i$ to be swapped with label $j$. In what follows, we consider three possible strategies for building the transition matrix $T$. (1) `Confusion matrix` (Algan & Ulusoy, 2020): we define $T$ as the oracle classifier's confusion matrix, up to a proper normalization to ensure the total flipping probability is $\epsilon$. We provide a theoretical study of this case in Appendix A1.3. (2) `Rare to most frequent class` (Xu et al., 2019): here, we flip the labels of the least frequent class with those of the most frequent class. This noise model is not uncommon in medical applications: imagine a setting where only a small fraction of the observations are abnormal, and thus likely to be annotated as the normal class. If switching between the rare and most frequent labels does not lead to a total probability of $\epsilon$, we move to the next least common class, and so on.

To set the stage for the experiments, we generate a synthetic data with $K = 10$ classes as follows. The features $X \in \mathbb{R}^d$ follow a standard multivariate Gaussian distribution of dimension $d = 100$. The conditional distribution of $Y \mid X$ is multinomial with weights $w_j(x) = \exp((x^\top B)_j)/\sum_{i=1}^{K} \exp((x^\top B)_i)$, where $B \in \mathbb{R}^{d \times K}$ whose entries are sampled independently from the standard normal distribution. In our experiments, we generate a total of $60,000$ data points, where $50,000$ are used to fit a classifier, and the remaining ones are randomly split to form calibration and test sets, each of size $5,000$. The training and calibration data are corrupted using the label noise models we defined earlier, with a fixed flipping probability of $\epsilon = 0.05$. Of course, the test set is not corrupted and contains the ground truth labels. We apply conformal prediction using both the HPS and the APS score functions, with a target coverage level $1 - \alpha$ of $90\%$. We use two predictive models: a two-layer neural network and an oracle classifier that has access to the conditional distribution of $\tilde{Y} \mid X$. Finally, we report the distribution of the coverage rate as in (2) and the prediction set sizes across 100 random splits of the calibration and test data. As a point of reference, we repeat the same experimental protocol described above on clean data; in this case, we do not violate the i.i.d. assumption required to grant the marginal coverage guarantee in (1).

The results are depicted in Figure 2. As expected, in the clean setting all conformal methods achieve the desired coverage of $90\%$. Under the `uniform flip` noise model, the coverage of the oracle classifier increases to around $94\%$, supporting our theoretical results from Section 2.1. The neural network model follows a similar trend. Although not supported by a theoretical guarantee, when corrupting the labels using the more challenging `confusion matrix` noise, we can see a conservative behavior similar to the `uniform flip`. By contrast, under the `rare to most frequent class` noise model, we can see a decrease in coverage, which is in line with our dis-

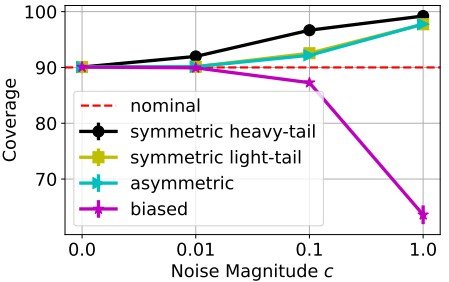 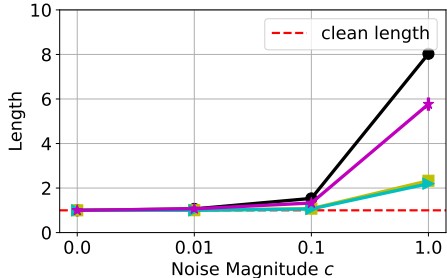

Figure 3: **Response-independent noise.** Performance of conformal prediction intervals with target coverage $1 - \alpha = 90\%$, using a noisy training set and a noisy calibration set. Left: Marginal coverage; Right: Length of predicted intervals (divided by the average clean length) using symmetric, asymmetric and biased noise with a varying magnitude. The results are evaluated over 50 independent experiments, with error bars showing one standard deviation. The standard deviation of $Y$ (without noise) and the square root of $\mathbb{E}\left[\mathrm{Var}\left[Y \mid X\right]\right]$ are both approximately 2.6.

claimer from Section 2.4. Yet, observe how the APS score tends to be more robust to label noise than HPS, which emphasizes the role of the score function.

In Appendix A2.1 we provide additional experiments with adversarial noise models that more aggressively reduce the coverage rate. Such adversarial cases are more pathological and less likely to occur in real-world settings, unless facing a malicious attacker.

## 3.2 REGRESSION

Similarly to the classification experiments, we study two types of noise distributions.

**Response-independent noise.** We consider an additive noise of the form: $\tilde{Y} = Y + c \cdot Z$, where $c$ is a parameter that allows us to control the noise level. The noise component $Z$ is a random variable sampled from the following distributions. (1) `Symmetric light tailed`: standard normal distribution; (2) `Symmetric heavy tailed`: t-distribution with one degree of freedom; (3) `Asymmetric`: standard gumbel distribution, normalized to have zero mean and unit variance; and (4) `Biased`: positive noise formulated as the absolute value of #2 above.

**Response-dependent noise.** Analogously to the class-dependent noise from Section 3.1, we define more challenging noise models as follows. (1) `Contractive`: this corruption pushes the ground truth response variables towards their mean. Formally, $\tilde{Y}_i = Y_i - \left(Y_i - \frac{1}{n}\sum_{i=1}^{n} Y_i\right) \cdot U$, where $U$ is a random uniform variable defined on the segment [0,0.5], and $n$ is the number of calibration points. (2) `Dispersive`: this noise introduces some form of a dispersion effect on the ground truth response, which takes the opposite form of the `contractive` model, given by $\tilde{Y}_i = Y_i + \left(Y_i - \frac{1}{n}\sum_{i=1}^{n} Y_i\right) \cdot U$.

Having defined the noise models, we turn to describe the data generating process. We simulate a 100-dimensional $X$ whose entries are sampled independently from a uniform distribution on the segment $[0, 5]$. Following Romano et al. (2019), the response variable is generated as follows:

$$Y \sim \mathrm{Pois}(\sin^2(\bar{X}) + 0.1) + 0.03 \cdot \bar{X} \cdot \eta_1 + 25 \cdot \eta_2 \cdot \mathbb{1}\left\{U < 0.01\right\}, \qquad (4)$$

where $\bar{X}$ is the mean of the vector $X$, and $\mathrm{Pois}(\lambda)$ is the Poisson distribution with mean $\lambda$. Both $\eta_1$ and $\eta_2$ are i.i.d. standard Gaussian variables, and $U$ is a uniform random variable on $[0, 1]$. The right-most term in (4) creates a few but large outliers. Figure A2 in the appendix illustrates the effect of the noise models discussed earlier on data sampled from (4).

We apply conformal prediction with the CQR score for each noise model as follows.[1] First, we fit a quantile random forest model on $8,000$ noisy training points; we then calibrate the model

---

[1]In our experiments, we use the original score from Romano et al. (2019) that can have negative values in contrast to the score defined in (3) as the former is more likely to be used in practice. Formally, the score we apply is $s^{\mathrm{CQR}}(x, y) := \max\{\hat{q}_{\alpha_{lo}}(x) - y, y - \hat{q}_{\alpha_{hi}}(x)\}$.

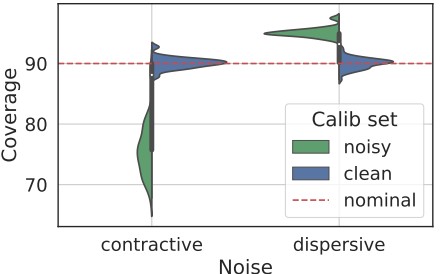 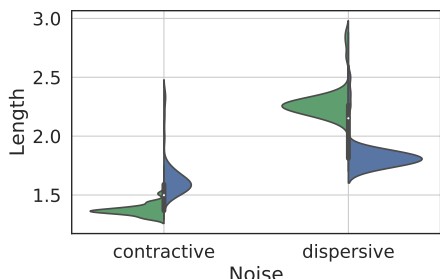

Figure 4: **Dispersive versus contractive noise regression experiment.** Performance of conformal prediction intervals with target coverage $1 - \alpha = 90\%$, using a noisy training set and a noisy calibration set. Left: Marginal coverage; Right: Length of predicted intervals. The results are evaluated over 50 independent experiments and the gray bar represents the interquartile range.

using $2,000$ fresh noisy samples; and, lastly, test the performance on additional $5,000$ clean, ground truth samples. The results are summarized in Figures 3 and 4. Observe how the prediction intervals tend to be conservative under `symmetric`, both for light- and heavy-tailed noise distributions, `asymmetric`, and `dispersive` corruption models. Intuitively, this is because these noise models increase the variability of $Y$; in Example 3 we prove this formally for any `symmetric` independent noise model, whereas here we show this result holds more generally even for response-dependent noise. By contrast, the prediction intervals constructed under the `biased` and `contractive` corruption models tend to under-cover the response variable. This should not surprise us: following Figure A2(c), the `biased` noise shifts the data 'upwards', and, consequently, the prediction intervals are undesirably pushed towards the positive quadrants. Analogously, the `contractive` corruption model pushes the data towards the mean, leading to intervals that are too narrow. Figure A5 in the appendix illustrates the scores achieved when using the different noise models and the 90%'th empirical quantile of the CQR scores. This figure supports the behaviour we see in Figures A3, 3 and A4: over-coverage is achieved when $\hat{q}_{\text{noisy}}$ is larger than $\hat{q}_{\text{clean}}$, and under-coverage is obtained when $\hat{q}_{\text{noisy}}$ is smaller.

In Section A2.2 of the Appendix we study the effect of the predictive model on the coverage property, for all noise models. To this end, we repeat similar experiments to the ones presented above, however, we now fit the predictive model on clean training data; the calibration data remains noisy. We also provide an additional adversarial noise model that reduces the coverage rate, but is unlikely to appear in real-world settings. Figures A3 and A4 in the appendix depict a similar behaviour for most noise models, except the `biased` noise for which the coverage requirement is not violated. This can be explained by the improved estimation of the low and high conditional quantiles, as these are fitted on clean data and thus less biased.

## 4 REAL DATA EXPERIMENTS

### 4.1 REGRESSION: AESTHETIC VISUAL RATING

In this section, we present a real-world application with continuous response, using Aesthetic Visual Analysis (AVA) data set, first presented by Murray et al. (2012). This data set contains pairs of images and their aesthetic scores in the range of 1 to 10, obtained by approximately 200 annotators. Following Kao et al. (2015); Talebi & Milanfar (2018); Murray et al. (2012), the task is to predict the average aesthetic score of a given test image. Therefore, we consider the average aesthetic score taken over all annotators as the clean, ground truth response. The noisy response is the average aesthetic score taken over 10 randomly selected annotators only.

We examine the performance of conformal prediction using both CQR and the *residual magnitude score*—the score from Section 2.3 with $\hat{u}(x) = 1$. We follow Talebi & Milanfar (2018) and take a transfer learning approach to fit the predictive model. Specifically, for feature extraction, we use a VGG-16 model—pre-trained on the ImageNet data set—whose last (deepest) fully connected layer is removed. Then, we feed the output of the VGG-16 model to a linear fully connected layer to predict the response. We trained two different models: a quantile regression model for CQR and a

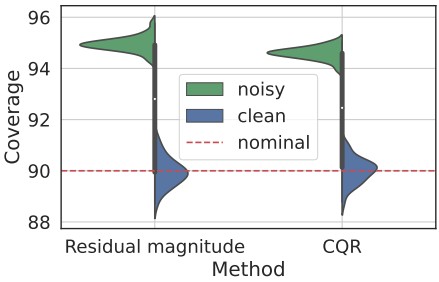 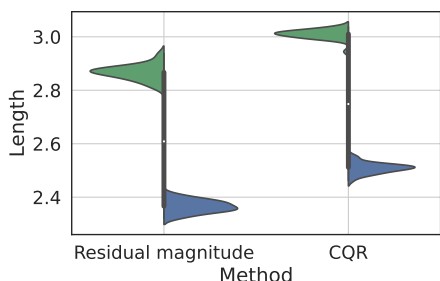

Figure 5: **Results for real-data regression experiment: predicting aesthetic visual rating.** Performance of conformal prediction intervals with 90% marginal coverage based on a VGG-16 model using a noisy training set. We compare the residual magnitude score and CQR methods with both noisy and clean calibration sets. Left: Marginal coverage; Right: Interval length. The results are evaluated over 30 independent experiments and the gray bar represents the interquartile range.

classic regression model for conformal with residual magnitude score. The models are trained on $34,000$ noisy samples, calibrated on $7,778$ noisy holdout points, and tested on $7,778$ clean samples. Further details regarding the training strategy are in Appendix A2.3.

Figure 5 portrays the marginal coverage and average interval length achieved using CQR and residual magnitude scores. As a point of reference, this figure also presents the performance of the two conformal methods when calibrated with a clean calibration set; as expected, the two perfectly attain 90% coverage. By constant, when calibrating the same predictive models with a noisy calibration set, the resulting prediction intervals tend to be wider and to over-cover the average aesthetic scores.

## 4.2 Classification: object recognition

We return to the classification experiment in Section 1. The CIFAR-10H data set contains the same 10,000 images as CIFAR-10, but with labels from a single annotator instead of a majority vote of 50 annotators. We fine-tune a ResNet18 model pre-trained on the clean training set of CIFAR-10, which contains 50,000 samples. Then we randomly select 2,000 observations from CIFAR-10H for calibration. The test set contains the remaining 8,000 samples, but with CIFAR-10 labels.

Figure 1 illustrates results obtained by applying conformal prediction with the APS score. We can see that (i) we obtain the exact desired coverage when using the clean calibration set; and (ii) when calibrating on noisy data, the constructed prediction sets over-cover the clean test labels. This is because the sets tend to be larger when calibrating on noisy data, as seen in Figure A6.

## 5 Discussion

**Related work.** Conformal prediction was first proposed by Vladimir Vovk and collaborators (Vovk et al., 1999; 2005). Recently there has been a body of work studying the statistical properties of conformal prediction (Lei et al., 2018; Barber, 2020) and its performance under deviations from exchangeability (Tibshirani et al., 2019; Podkopaev & Ramdas, 2021; Barber et al., 2022). Label noise independently of conformal prediction has been well-studied; see, for example Angluin & Laird (1988); Tanno et al. (2019); Frénay & Verleysen (2013). To our knowledge, conformal prediction under label noise has not been previously analyzed. The closest work to ours is that of Cauchois et al. (2022) studying conformal prediction with weak supervision, which could be interpreted as a type of noisy label.

**Future directions.** Our work raises many new questions. First, one can try and define a score function that is more robust to label noise, continuing the line of Gendler et al. (2021); Frénay & Verleysen (2013); Cheng et al. (2022). Second, an important remaining question is whether or not conformal risk control (Angelopoulos et al., 2021; 2022) more generally is also robust to label noise. Lastly, it would be interesting to analyze the robustness of alternative conformal methods such as cross-conformal and jackknife+ (Vovk, 2015; Barber et al., 2021) that do not require data-splitting.

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

# Appendices

## A1    MATHEMATICAL PROOFS

*Theorem 1.* Our assumption states that

$$\mathbb{P}(\tilde{s}_{\text{test}} \leq t) \leq \mathbb{P}(s_{\text{test}} \leq t).$$

Note that the probability is only taken over $\tilde{s}_{\text{test}}$. Since $\hat{q}_{\text{noisy}}$ is constant (measurable) with respect to this probability, we have that, for any $\alpha \in (0, 1)$,

$$\mathbb{P}(s_{\text{test}} \leq \hat{q}_{\text{noisy}}) \geq \mathbb{P}(\tilde{s}_{\text{test}} \leq \hat{q}_{\text{noisy}}) \geq 1 - \alpha.$$

This implies that $Y_{\text{test}} \in \widehat{\mathcal{C}}_{\text{noisy}}(X_{\text{test}})$ with probability at least $1 - \alpha$, completing the proof of the lower bound.

Regarding the upper bound, by the same argument,

$$\mathbb{P}(s_{\text{test}} \leq \hat{q}_{\text{noisy}}) \leq \mathbb{P}(\tilde{s}_{\text{test}} \leq \hat{q}_{\text{noisy}}) + u \leq 1 - \alpha + \frac{1}{n+1} + u.$$

$\square$

### A1.1    APS SCORE

**Lemma A1.** *Let $\hat{f}_y(x) = \hat{f}(Y = y \mid X = x)$ be any classifier that ranks the classes in the same order as the oracle model, i.e., one satisfying, for all $y \neq y'$,*

$$\hat{f}_y(x) \geq \hat{f}_{y'}(x) \iff \hat{\pi}_y(x) \geq \hat{\pi}_{y'}(x). \tag{A1}$$

*Let $s$ be the (deterministic) APS conformal score with model $\hat{f}$,*

$$s(x, y) = \sum_{y' \in \mathcal{Y}} \hat{f}_{y'}(x) \mathbb{1}\left\{\hat{f}_{y'}(x) \geq \hat{f}_y(x)_y\right\}.$$

*Consider the corruption function $g^{\text{flip}}$. Then,*

$$\mathbb{P}(s(X, Y) \leq t) \geq \mathbb{P}(s(X, \tilde{Y}) \leq t).$$

*Proof of Lemma A1.* For notational convenience, assume the classes are ordered such that $\hat{f}_y(x)$ is decreasing in $y$. Fix $x \in \mathcal{X}$, let $p_k = \mathbb{P}(Y = k \mid X = x)$ and let us use the shorthand $\hat{\pi}_k = \mathbb{P}(\tilde{Y} = k \mid X = x)$. Finally, draw $(X, Y, \tilde{Y})$ from their joint distribution $\mathbb{P}$.

Notice that in this context, the assumption (A1) simply states that $\hat{\pi}_k$ is sorted from greatest to least. Furthermore, by the definition of $g^{\text{flip}}$,

$$\hat{\pi}_k = p_k(1 - (K - 1)\epsilon/K) + (1 - p_k)\epsilon/K = p_k(1 - \epsilon) + \epsilon/K. \tag{A2}$$

This implies that $\hat{\pi}_k$ and $p_k$ have the same ordering, and therefore, that the entries $p_k$ are also decreasing in $k$. In the ensuing argument, this fact will be crucial.

Starting with the law of total probability,

$$\mathbb{P}(s(x, \tilde{Y}) \leq t \mid X = x) = \sum_{k \in \mathcal{Y}} \hat{\pi}_k \mathbb{1}\{s(x, k) \leq t\} = \sum_{k: s(x, k) \leq t} \hat{\pi}_k. \tag{A3}$$

Take $k^* = |\{k : s(x, k) \leq t\}|$; if $k^* = 0$, the proof becomes trivial by noticing that $\mathbb{P}(s(x, \tilde{Y}) \leq t \mid X = x) = 0 \leq \mathbb{P}(s(x, Y) \leq t \mid X = x)$ and applying the tower property. Therefore, we continue the proof in the case that $k^* \geq 1$. Because $s(x, k)$ is monotone in $k$, we can rewrite the last expression as

$$\sum_{k: s(x, k) \leq t} \hat{\pi}_k = \sum_{k=1}^{k^*} \hat{\pi}_k.$$

Applying (A2) and using the fact that $p_k$ is sorted in decreasing order yields

$$\sum_{k=1}^{k^*} \hat{\pi}_k = (1-\epsilon) \sum_{k=1}^{k^*} p_k + k^* \epsilon / K = \sum_{k=1}^{k^*} p_k + \epsilon \left( k^*/K - \sum_{k=1}^{k^*} p_k \right).$$

Next we will critically use the fact that $p$ is sorted from greatest to least. This sorting implies that $(p_1 + \ldots + p_k) \geq k/K$ for all $k$ (if this failed at any $k$, we would have $(p_1 + \ldots + p_K) < 1$, which is impossible). Applying this fact at the $k^*$th index,

$$\sum_{k=1}^{k^*} p_k + \epsilon \left( k^*/K - \sum_{k=1}^{k^*} p_k \right) \leq \sum_{k=1}^{k^*} p_k, \tag{A4}$$

as the term multiplied by $\epsilon$ is negative. Reversing the steps from (A3), one can rewrite the limits of the sum as an indicator function, which by law of total probability yields the CDF of the scores,

$$\sum_{k=1}^{k^*} p_k = \sum_{k:s(x,k)\leq t} p_k = \sum_{k\in\mathcal{Y}} p_k \mathbb{1}\left\{s(X,k) \leq t\right\} = \mathbb{P}(s(x,Y) \leq t \mid X = x). \tag{A5}$$

We have so far proved that

$$\mathbb{P}(s(x,\tilde{Y}) \leq t \mid X = x) \leq \mathbb{P}(s(x,Y) \leq t \mid X = x);$$

taking expectations on both sides yields the conclusion,

$$\mathbb{P}(s(X,\tilde{Y}) \leq t) \leq \mathbb{P}(s(X,Y) \leq t).$$

$\square$

*Proof of Example 1.* To complete the proof of the lower bound it suffices to show that $\mathbb{P}(s(X,Y) \leq t) \geq \mathbb{P}(s(X,\tilde{Y}) \leq t)$. We prove this in Lemma A1.

For the coverage upper-bound, combining (A4) and (A5), witness that

$$\mathbb{P}(s(x,\tilde{Y}) \leq t \mid X = x) = \sum_{k=1}^{k^*} p_k + \epsilon \left( k^*/K - \sum_{k=1}^{k^*} p_k \right)$$

$$= \mathbb{P}(s(x,Y) \leq t \mid X = x) + \epsilon \left( k^*/K - \sum_{k=1}^{k^*} p_k \right).$$

We can lower-bound this expression by realizing that $k^*/K - \sum_{k=1}^{k^*} p_k$ takes values no lower than $-(K-1)/K$. Therefore,

$$\mathbb{P}(s(x,\tilde{Y}) \leq t \mid X = x) + \epsilon \frac{K-1}{K} \geq \mathbb{P}(s(x,Y) \leq t \mid X = x).$$

Applying the tower property and noting that by the standard conformal argument Vovk et al. (2005), gives $\mathbb{P}(s(x,\tilde{Y}) \leq t) \geq 1 - \alpha + 1/(n+1)$ yields

$$1 - \alpha + \frac{1}{n+1} + \epsilon \frac{K-1}{K} \geq \mathbb{P}(s(x,Y) \leq t).$$

$\square$

### A1.2 HPS score

*Proof of Example 2.* For proof with the HPS function, we consider the following corruption model:

$$g^{\text{flip}+}(y) = \begin{cases} y & \text{w.p } 1-\epsilon \\ Y' & \text{else,} \end{cases} \tag{A6}$$

where $Y'$ is uniformly drawn from the set $\{1, \ldots, K\} \setminus Y$. This is the same as $g^{\text{flip}}$, but rescaled so that $\epsilon$ in $g^{\text{flip}}$ corresponds to $\epsilon(K-1)/K$ in $g^{\text{flip}+}$.

Consider the corrupted oracle nonconformity score $s(x, y) = 1 - \mathbb{P}(\tilde{Y} \mid X)$. By definition of this score function, we can explicitly write the probability mass functions of $\tilde{s}_i$ and $s_i$ as follows:

$$\mathbb{P}\left(\tilde{s}_i = t \mid X\right) = \sum_{j=1}^{K} \mathbb{P}(\tilde{Y} = j \mid X) \mathbb{1}\left\{t = 1 - \mathbb{P}(\tilde{Y} = j \mid X)\right\} \text{ and}$$

$$\mathbb{P}\left(s_i = t \mid X\right) = \sum_{j=1}^{K} \mathbb{P}(Y = j \mid X) \mathbb{1}\left\{t = 1 - \mathbb{P}(\tilde{Y} = j \mid X)\right\}.$$

Note that these two distributions are supported on the same discrete set, $\mathcal{D} = \left\{1 - \mathbb{P}(\tilde{Y} = j | X), j = 1, ..., K\right\}$. Applying the definition of our corruption model and rearranging yields

$$\mathbb{P}(\tilde{Y} = j \mid X) = (1 - \frac{K}{K-1}\epsilon)\mathbb{P}(Y = j \mid X) + \frac{\epsilon}{K-1}. \tag{A7}$$

Using this fact and the law of total probability, we can write

$$\mathbb{P}\left(s_i \leq q\right) - \mathbb{P}\left(\tilde{s}_i \leq q\right) = \sum_{\substack{t \in \mathcal{D} \\ t \leq q}} \sum_{j=1}^{K} \left(\mathbb{P}(Y = j \mid X) - \mathbb{P}(\tilde{Y} = j \mid X)\right) \mathbb{1}\left\{t = 1 - \mathbb{P}(\tilde{Y} = j \mid X)\right\} \tag{A8}$$

$$= \frac{\epsilon K}{K-1} \sum_{\substack{t \in \mathcal{D} \\ t \leq q}} \sum_{j=1}^{K} \left(\mathbb{P}(Y = j \mid X) - \frac{1}{K}\right) \mathbb{1}\left\{\mathbb{P}(Y = j \mid X) = \frac{1 - \frac{\epsilon}{K-1} - t}{1 - \frac{K}{K-1}\epsilon}\right\}$$

$$= \frac{\epsilon K}{K-1} \sum_{\substack{t \in \mathcal{D} \\ t \leq q}} \left(\frac{1 - \frac{\epsilon}{K-1} - t}{1 - \frac{K}{K-1}\epsilon} - \frac{1}{K}\right),$$

where the last step holds because each summand is nonzero at exactly one value of $j$.

To prove the lower bound, we will show that $\mathbb{P}\left(s_i \leq q\right) - \mathbb{P}\left(\tilde{s}_i \leq q\right) \geq 0$. It suffices to show that each summand in the last line of (A8) is positive. This is trivially true when $t > 1 - \epsilon/(K-1)$ because by (A7), we have that $\inf_j \mathbb{P}(\tilde{Y} = j \mid X) \geq \epsilon/(K-1)$, and thus $q = 1$. To complete the proof, our goal is to show that if $t \leq 1 - \epsilon/(K-1)$ and $\epsilon \leq 1 - 1/K$, then

$$\frac{1 - \frac{\epsilon}{K-1} - t}{1 - \frac{K}{K-1}\epsilon} \geq \frac{1}{K}.$$

Rearranging terms in the above expression gives

$$1 - \frac{1}{K} \geq t.$$

This is implied directly by combining the two bounds $t \leq 1 - \epsilon/(K-1)$ and $\epsilon \leq 1 - 1/K$. This completes the proof of the lower bound.

To prove the upper bound, we need to show that

$$P(s_i \leq q) - \mathbb{P}(\tilde{s}_i \leq q) \leq \frac{\epsilon K \min(\lfloor 1/(1-q) \rfloor, K)}{K-1}\left(\frac{1 - \frac{\epsilon}{K-1}}{1 - \frac{K}{K-1}\epsilon} - 1\right).$$

The last line of (A8) can be rewritten as

$$\frac{\epsilon K}{K-1} \sum_{\substack{t \in \mathcal{D} \\ t \leq q}} \left(\frac{1 - \frac{\epsilon}{K-1} - t}{1 - \frac{K}{K-1}\epsilon} - \frac{1}{K}\right) = \frac{\epsilon K}{K-1} |\{t \in \mathcal{D} : t \leq q\}| \left(\frac{1 - \frac{\epsilon}{K-1}}{1 - \frac{K}{K-1}\epsilon} - \frac{1}{K}\right) - \frac{\epsilon K}{K-1} \sum_{\substack{t \in \mathcal{D} \\ t \leq q}} t$$

$$\leq \frac{\epsilon K^2}{K-1} \left(\frac{1 - \frac{\epsilon}{K-1}}{1 - \frac{K}{K-1}\epsilon} - 1\right) = \frac{\epsilon^2 \frac{K}{K-1}}{1 - \frac{K}{K-1}\epsilon} K.$$

where the inequality upper-bounded the cardinality by $K$ and the last term by 0.

To recover the statement in the main paper, one can rewrite this final expression with $\epsilon(K-1)/K$ substituted in wherever the term $\epsilon$ appears. Rearranging terms then gives the result. $\qquad \square$

A1.3 CONFUSION MATRIX

The confusion matrix noise model is more realistic than the uniform flip. However, there exists a score function that causes conformal prediction to fail for any non-identity confusion matrix. We define the corruption model as follows: consider a matrix $T$ in which $(T)_{i,j} = \mathbb{P}(\tilde{Y} = j \mid Y = i)$.

$$
g^{\mathrm{confusion}}(y) = \begin{cases} 1 & \text{w.p. } T_{1,y} \\ \cdots & \\ K & \text{w.p. } T_{K,y}. \end{cases}
$$

**Example 5.** *Let $\widehat{C}_{\mathrm{noisy}}$ be constructed as in Recipe 1 with any score function $s$ and the corruption function $g^{\mathrm{confusion}}$. Then,*

$$
\mathbb{P}\left(Y_{\mathrm{test}} \in \widehat{C}_{\mathrm{noisy}}(X_{\mathrm{test}})\right) \geq 1 - \alpha.
$$

*if and only if for all classes $j \in \{1, \ldots, K\}$,*

$$
\sum_{j'=1}^{K} \mathbb{P}(\tilde{Y} = j' \mid Y = j)\mathbb{P}(\tilde{s} \leq t \mid \tilde{Y} = j') \geq \mathbb{P}(s \leq t \mid Y = j).
$$

The proof is below. In words, the necessary and sufficient condition is for the noise distribution/confusion matrix $\mathbb{P}(\tilde{Y} = j' \mid Y = j)$ to place sufficient mass on those classes $j'$ whose quantiles which are larger than $\mathbb{P}(s \leq t \mid Y = j)$. However, without assumptions on the model and score, the conditional probabilities are unknown, so it is impossible to say which noise distributions will preserve coverage.

*Proof.* By law of total probability, $\mathbb{P}(s \leq t) = \mathbb{E}\left[\mathbb{P}(s \leq t \mid Y = j)\right] = \sum_{j=1}^{K} w_j \mathbb{P}(s \leq t \mid Y = j)$. But under the noisy model, we have instead that

$$
\mathbb{P}(\tilde{s} \leq t) = \mathbb{E}\left[\mathbb{P}(\tilde{s} \leq t \mid \tilde{Y} = j')\right] = \sum_{j=1}^{K} \sum_{j'=1}^{K} w_j \mathbb{P}(\tilde{Y} = j' \mid Y = j)\mathbb{P}(\tilde{s} \leq t \mid \tilde{Y} = j').
$$

We can write

$$
\mathbb{P}(s \leq t) - \mathbb{P}(\tilde{s} \leq t) = \sum_{j=1}^{K} \sum_{j'=1}^{K} w_j \mathbb{P}(\tilde{Y} = j' \mid Y = j)\mathbb{P}(\tilde{s} \leq t \mid \tilde{Y} = j') - \sum_{j=1}^{K} w_j \mathbb{P}(s \leq t \mid Y = j).
$$

Combining the sums and factoring, the above display equals

$$
\sum_{j=1}^{K} w_j \left( \sum_{j'=1}^{K} \left( \mathbb{P}(\tilde{Y} = j' \mid Y = j)\mathbb{P}(\tilde{s} \leq t \mid \tilde{Y} = j') \right) - \mathbb{P}(s \leq t \mid Y = j) \right).
$$

We can factor this expression as

$$
\sum_{j=1}^{K} w_j \mathbb{P}(s \leq t \mid Y = j) \sum_{j'=1}^{K} \left( \mathbb{P}(\tilde{Y} = j' \mid Y = j)\frac{\mathbb{P}(\tilde{s} \leq t \mid \tilde{Y} = j')}{\mathbb{P}(s \leq t \mid Y = j)} - \frac{1}{K} \right).
$$

The stochastic dominance condition holds uniformly over all choices of base probabilities $w_j$ if and only if for all $j \in [K]$,

$$
\sum_{j'=1}^{K} \mathbb{P}(\tilde{Y} = j' \mid Y = j)\mathbb{P}(\tilde{s} \leq t \mid \tilde{Y} = j') \geq \mathbb{P}(s \leq t \mid Y = j).
$$

$\square$

Notice that the left-hand side of the above display is a convex mixture of the quantiles $\mathbb{P}(\tilde{s} \leq t \mid \tilde{Y} = j')$ for $j' \in [K]$. Thus, the necessary and sufficient condition is for the noise distribution $\mathbb{P}(\tilde{Y} = j' \mid Y = j)$ to place sufficient mass on those classes $j'$ whose quantiles which are larger than $\mathbb{P}(s \leq t \mid Y = j)$. But of course, without assumptions on the model and score, the latter are unknown, so it is impossible to say which noise distributions will preserve coverage.

### A1.4 CQR SCORE

*Proof of Example 3.* We will show that $\mathbb{P}\left(\tilde{s} \leq q \mid X = x\right) \leq \mathbb{P}\left(s \leq q \mid X = x\right)$. Notice that we will only prove for the case where $q \geq 0$ since according to the score function, for $q \leq 0$ both probabilities are equal to zero. Formally,

$$
\begin{aligned}
\mathbb{P}\left(\tilde{s} \leq q \mid X = x\right) &= \mathbb{P}\left(\max\{\hat{q}_{\alpha_{lo}}\left(x\right) - Y - Z, Y + Z - \hat{q}_{\alpha_{hi}}\left(x\right)\} \leq q \mid X = x\right) \\
&= \mathbb{P}\left(\hat{q}_{\alpha_{lo}}\left(x\right) - Y - Z \leq q, Y + Z - \hat{q}_{\alpha_{hi}}\left(x\right) \leq q \mid X = x\right) \\
&= \mathbb{P}\left(\hat{q}_{\alpha_{lo}}\left(x\right) - q \leq Y + Z \leq \hat{q}_{\alpha_{hi}}\left(x\right) + q \mid X = x\right) \\
&= 1 - \mathbb{P}\left(Y + Z \geq \hat{q}_{\alpha_{hi}}\left(x\right) + q \mid X = x\right) - \mathbb{P}\left(Y + Z \leq \hat{q}_{\alpha_{lo}}\left(x\right) - q \mid X = x\right).
\end{aligned}
$$

Similarly,

$$
\mathbb{P}\left(s \leq q \mid X = x\right) = 1 - \mathbb{P}\left(Y \geq \hat{q}_{\alpha_{hi}}\left(x\right) + q \mid X = x\right) - \mathbb{P}\left(Y \leq \hat{q}_{\alpha_{lo}}\left(x\right) - q \mid X = x\right).
$$

According to Lemma A2,

$$
\begin{aligned}
&1 - \mathbb{P}\left(Y + Z \geq \hat{q}_{\alpha_{hi}}\left(x\right) + q \mid X = x\right) - \mathbb{P}\left(Y + Z \leq \hat{q}_{\alpha_{lo}}\left(x\right) - q \mid X = x\right) \\
&\leq 1 - \mathbb{P}\left(Y \geq \hat{q}_{\alpha_{hi}}\left(x\right) + q \mid X = x\right) - \mathbb{P}\left(Y \leq \hat{q}_{\alpha_{lo}}\left(x\right) - q \mid X = x\right),
\end{aligned}
$$

under the assumption that

$$
\hat{q}_{\alpha_{lo}}\left(x\right) \leq q_{1/2}(x) \leq \hat{q}_{\alpha_{hi}}\left(x\right),
$$

where $q_{1/2}(x)$ is the true median of $\tilde{Y} \mid X = x$. By the law of total probability this inequality holds also without conditioning on $X$. $\qquad\square$

**Lemma A2.** *Suppose $Y$ has a density that is unimodal and symmetric around $0$ and $Z$ is symmetric, i.e., $Z$ has the same distribution as $-Z$. Suppose further that $Y$ and $Z$ are independent. Then for any $q > 0$,*

$$
\mathbb{P}(Y + Z \leq -q) \geq \mathbb{P}(Y \leq -q)
$$

*and*

$$
\mathbb{P}(Y + Z \geq q) \geq \mathbb{P}(Y \geq q).
$$

*Proof of Lemma A2.*

$$
\begin{aligned}
\mathbb{P}(Y + Z \leq -q) &= \mathbb{E}\left[\mathbb{P}(Y + Z \leq -q \mid |Z| = z)\right] \\
&= \mathbb{E}\left[\frac{1}{2}\mathbb{P}(Y \leq -q - z) + \frac{1}{2}\mathbb{P}(Y \leq -q + z)\right] \\
&= \mathbb{P}(Y \leq -q) + \mathbb{E}\left[\frac{1}{2}\left(\mathbb{P}(Y \leq -q - z) - \mathbb{P}(Y \leq -q)\right) + \frac{1}{2}\left(\mathbb{P}(Y \leq -q + z) - \mathbb{P}(Y \leq -q)\right)\right] \\
&= \mathbb{P}(Y \leq -q) + \mathbb{E}\left[\frac{1}{2}\left(\mathbb{P}(Y \leq -q + z) - \mathbb{P}(Y \leq -q)\right) - \frac{1}{2}\left(\mathbb{P}(Y \leq -q) - \mathbb{P}(Y \leq -q - z)\right)\right] \\
&= \mathbb{P}(Y \leq -q) + \mathbb{E}\left[\underbrace{\frac{1}{2}\mathbb{P}(-q < Y \leq -q + z) - \frac{1}{2}\mathbb{P}(-q - z < Y \leq -q)}_{\geq 0}\right] \\
&\geq \mathbb{P}(Y \leq -q).
\end{aligned}
$$

Above, the expectations are over $z \geq 0$ which we take to be a random variable with the same distribution as $|Z|$. The final inequality follows from the fact that the density of $Y$ is unimodal.

The proof for the upper tail is similar and we omit it. $\qquad\square$

A1.5    Normalized residual magnitude score

*Proof of Example 4.* Equation 2.3 directly implies

$$\mathbb{P}\Big(\big|Y - \mathbb{E}[Y \mid X = x]\big|/\hat{u}(x) > t \,\Big|\, X = x\Big) \le \mathbb{P}\Big(\big|Y + Z - \mathbb{E}[Y \mid X = x]\big|/\hat{u}(x) > t \,\Big|\, X = x\Big),$$

since $\hat{u}(x) > 0$ and both probabilities are $X$-conditional. Furthermore, since we are using the oracle model $\hat{f} = \mathbb{E}[Y \mid X = x]$, this is identical to the statement

$$\mathbb{P}(s_{\text{test}}^{\text{RM}} \le t) \ge \mathbb{P}(\tilde{s}_{\text{test}}^{\text{RM}} \le t).$$

The above statement says that the clean score distribution is stochastically dominated by the noisy score distribution. Since $\hat{q}_{\text{noisy}}$ is independent of $s_{\text{test}}^{\text{RM}}$ and $\tilde{s}_{\text{test}}^{\text{RM}}$, we know

$$\mathbb{P}(s_{\text{test}}^{\text{RM}} \le \hat{q}_{\text{noisy}}) \ge \mathbb{P}(\tilde{s}_{\text{test}}^{\text{RM}} \le \hat{q}_{\text{noisy}}) \ge 1 - \alpha.$$

This completes the argument.    □

The following is a direct corollary of Example 4.

**Corollary 3.** *Suppose the $\tilde{Y}_i$ are drawn I.I.D. $\tilde{Y}_i = Y_i + Z_i$. Assume there exist constants $c$ and $C$ such that*

$$\mathbb{P}(|Y_i - \mathbb{E}[Y_i|X_i]| \le t) \ge 1 - c\exp(-Ct^2) \text{ and } \mathbb{P}(|Y_i + Z_i - \mathbb{E}[Y_i|X_i]| \le t) \le 1 - c\exp(-Ct^2).$$

*Then*

$$\mathbb{P}(s_{n+1} \le \hat{q}) \ge 1 - \alpha.$$

There is nothing special about the sub-Gaussian tail decay in the above corollary; any tail decay conditions will work if they can be chained together.

A1.6    Distribution-free results

*Proof of Proposition 1.* For convenience, assume the existence of probability density functions $\tilde{p}$ and $p$ for $\tilde{Y}$ and $Y$ respectively (these can be taken to be probability mass functions if $\mathcal{Y}$ is discrete). Also define the multiset of $Y$ values $E = \{Y_1, ..., Y_n\}$ and the corresponding multiset of $\tilde{Y}$ values $\tilde{E}$. Take the set

$$\mathcal{A} = \{y : \tilde{p}(y) > p(y)\}.$$

Since $Y \overset{d}{\ne} \tilde{Y}$, we know that the set $\mathcal{A}$ is nonempty and $\mathbb{P}(\tilde{Y} \in \mathcal{A}) = \delta_1 > \mathbb{P}(Y \in \mathcal{A}) = \delta_2 \ge 0$. The adversarial choice of score function will be $s(x, y) = \mathbb{1}\{y \in \mathcal{A}^c\}$; it puts high mass wherever the ground truth label is more likely than the noisy label. The crux of the argument is that this design makes the quantile smaller when it is computed on the noisy data than when it is computed on clean data, as we next show.

Begin by noticing that, because $s(x, y)$ is binary, $\hat{q}_{\text{clean}}$ is also binary, and therefore $\hat{q}_{\text{clean}} > t \iff \hat{q}_{\text{clean}} = 1$. Furthermore, $\hat{q}_{\text{clean}} = 1$ if and only if $|E \cap \mathcal{A}| < \lceil(n+1)(1-\alpha)\rceil$. Thus, these events are the same, and for any $t \in (0, 1]$,

$$\mathbb{P}(\hat{q}_{\text{clean}} \ge t) = \mathbb{P}\Big(\big|E \cap \mathcal{A}\big| < \lceil(n+1)(1-\alpha)\rceil\Big).$$

By the definition of $\mathcal{A}$, we have that $\mathbb{P}\Big(|E \cap \mathcal{A}| < \lceil(n+1)(1-\alpha)\rceil\Big) > \mathbb{P}\Big(|\tilde{E} \cap \mathcal{A}| < \lceil(n+1)(1-\alpha)\rceil\Big)$. Chaining the inequalities, we get

$$\mathbb{P}(\hat{q}_{\text{clean}} \ge t) > \mathbb{P}\Big(|\tilde{E} \cap \mathcal{A}| < \lceil(n+1)(1-\alpha)\rceil\Big) = \mathbb{P}(\hat{q} \ge t).$$

Since $s_{n+1}$ is measurable with respect to $E$ and $\tilde{E}$, we can plug it in for $t$, yielding the conclusion.    □

**Remark 1.** *In the above argument, if one further assumes continuity of the (ground truth) score function and $\mathbb{P}(\tilde{Y} \in \mathcal{A}) = \mathbb{P}(Y \in \mathcal{A}) + \rho$ for*

$$\rho = \inf \left\{ \rho' > 0 \ : \text{BinomCDF}(n, \delta_1, \lceil (n+1)(1-\alpha) \rceil - 1) + \frac{1}{n} < \right.$$
$$\left. \text{BinomCDF}(n, \delta_2 + \rho', \lceil (n+1)(1-\alpha) \rceil - 1) \right\},$$

*then*

$$\mathbb{P}(s_{n+1} \leq \hat{q}) < 1 - \alpha.$$

In other words, the noise must have some sufficient magnitude in order to disrupt coverage.

*Proof of Corollary 1.* This a consequence of the TV bound from Barber et al. (2022) with weights identically equal to 1. ∎

Unfortunately, getting such a TV bound requires a case by case analysis. It's not even straightforward to get a TV bound under strong Gaussian assumptions.

**Proposition 2** (No general TV bound). *Assume $Y \sim \mathcal{N}(0, \tau^2)$ and $\tilde{Y} = y + Z$, where $Z \sim \mathcal{N}(0, \sigma^2)$. Then $D_{\text{TV}}(Y, \tilde{Y}) \overset{\tau \to 0}{\to} 1$.*

*Proof.*

$$\text{TV}(\mathcal{N}(0, \tau^2), \mathcal{N}(0, \tau^2 + \sigma^2)) = \int_{-\infty}^{\infty} \left| e^{-x^2/\tau^2} - e^{-x^2/(\tau^2+\sigma^2)} \right| dx \overset{\tau \to 0}{\to} 1.$$

∎

## A2 ADDITIONAL EXPERIMENTAL DETAILS AND RESULTS

### A2.1 SYNTHETIC CLASSIFICATION: ADVERSARIAL NOISE MODELS

In contrast with the noise distributions presented in Section 3.1, here we construct adversarial noise models to intentionally reduce the coverage rate.

1. `Most frequent confusion`: we extract from the confusion matrix the pair of classes with the highest probability to be confused between each other, and switch their labels until reaching a total probability of $\epsilon$. In cases where switching between the most common pair is not enough to reach $\epsilon$, we proceed by flipping the labels of second most confused pairs of labels, and so on.

2. `Wrong to right`: wrong predictions during calibration cause larger predictions sets during test time. Hence making the model think it makes less mistakes then actual during calibration can lead to under-coverage during test time. Here, we first observe the model predictions over the calibration set, and then switch the labels only of points that were misclassified. We switch the label to the class that is most likely to be the correct class according to the model, hence making the model think it was correct. We switch a suitable amount of labels in order to reach a total switching probability of $\epsilon$ (this noise model assumes there are enough wrong predictions in order to do so).

3. `Optimal adversarial`: we describe here an algorithm for building the worst possible label noise for a specific model using a specific non-conformity score. This noise will decrease the calibration threshold at most and as a result, will cause significant under-coverage during test time. To do this, we perform an iterative process. In each iteration we calculate the non-conformity scores of all of the calibration points with their current labels. We calculate the calibration threshold as in regular conformal prediction and then, from the points that have a score above the threshold, we search for the one that switching its label can reduce its score by most. We switch the label of this point to the label that gives the lowest score and then repeat the iterative process with the new set of labels. Basically, at every step we make the label swap that will decrease the threshold by most.

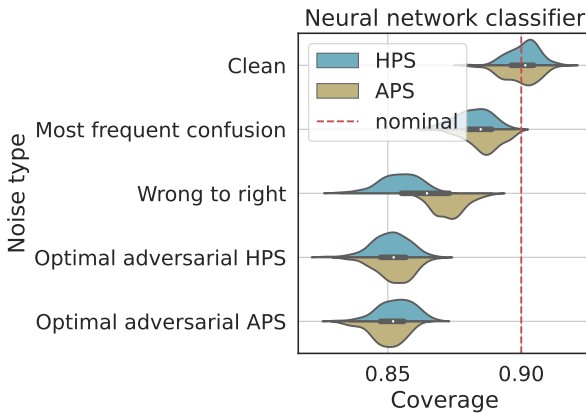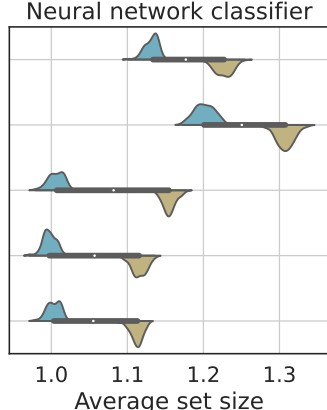

Figure A1: **Effect of label noise on synthetic multi-class classification data**. Performance of conformal prediction sets with target coverage $1 - \alpha = 90\%$, using a noisy training set and a noisy calibration set with adversarial noise models. Left: Marginal coverage; Right: Average size of predicted sets. The results are evaluated over 100 independent experiments.

In these experiments we apply the same settings as described in Section 3.1 of the main manuscript and present the results in Figure A1. We can see that the `optimal adversarial` noise causes the largest decrease in coverage as one would expect. The `most-frequent-confusion` noise decreases the neural network coverage to approximately $89\%$. The `wrong-to-right` noise decreases the coverage to around $85\%$ with the HPS score and to around $87\%$ with the APS score. This gap is expected as this noise directly reduces the HPS score. We can see that the optimal worst case noise for each score function reduces the coverage to around $85\%$ when using that score. This is in fact the maximal decrease in coverage possible theoretically, hence it strengthens the optimally of our iterative algorithm.

### A2.2 SYNTHETIC REGRESSION: ADDITIONAL RESULTS

Here we first illustrate in Figure A2 the data we generate in the synthetic regression experiment and the different corruptions we apply.

In Section 3.2 of the main manuscript we apply some realistic noise models and examine the performance of conformal prediction using CQR score with noisy training and calibration sets. Here we construct some more experiments using the same settings, however we train the models using clean data instead of noisy data. Moreover, we apply an additional adversarial noise model that differs from those presented in Section 3.2 in the sense that it is designed to intentionally reduce the coverage level.

`Wrong to right`: an adversarial noise that depends on the underlying trained regression model. In order to construct the noisy calibration set we switch 7% of the responses as follows: we randomly swap between outputs that are not included in the interval predicted by the model and outputs that are included.

Figures A3 and A4 depict the marginal coverage and interval length achieved when applying the different noise models. We see that the adversarial `wrong to right` noise model reduces the coverage rate to approximately 83%. Moreover, these results are similar to those achieved in Section 3.2, except for the conservative coverage attained using `biased` noise, which can be explained by the more accurate low and high estimated quantiles.

Lastly, in order to explain the over-coverage or under-coverage achieved for some of the different noise models, as depicted in Figures 3 and 4, we present in Figure A5 the CQR scores and their 90%'th empirical quantile. Over-coverage is achieved when the noisy scores are larger than the clean ones, for example, in the `symmetric heavy tailed` case, and under-coverage is achieved when the noisy scores are smaller.

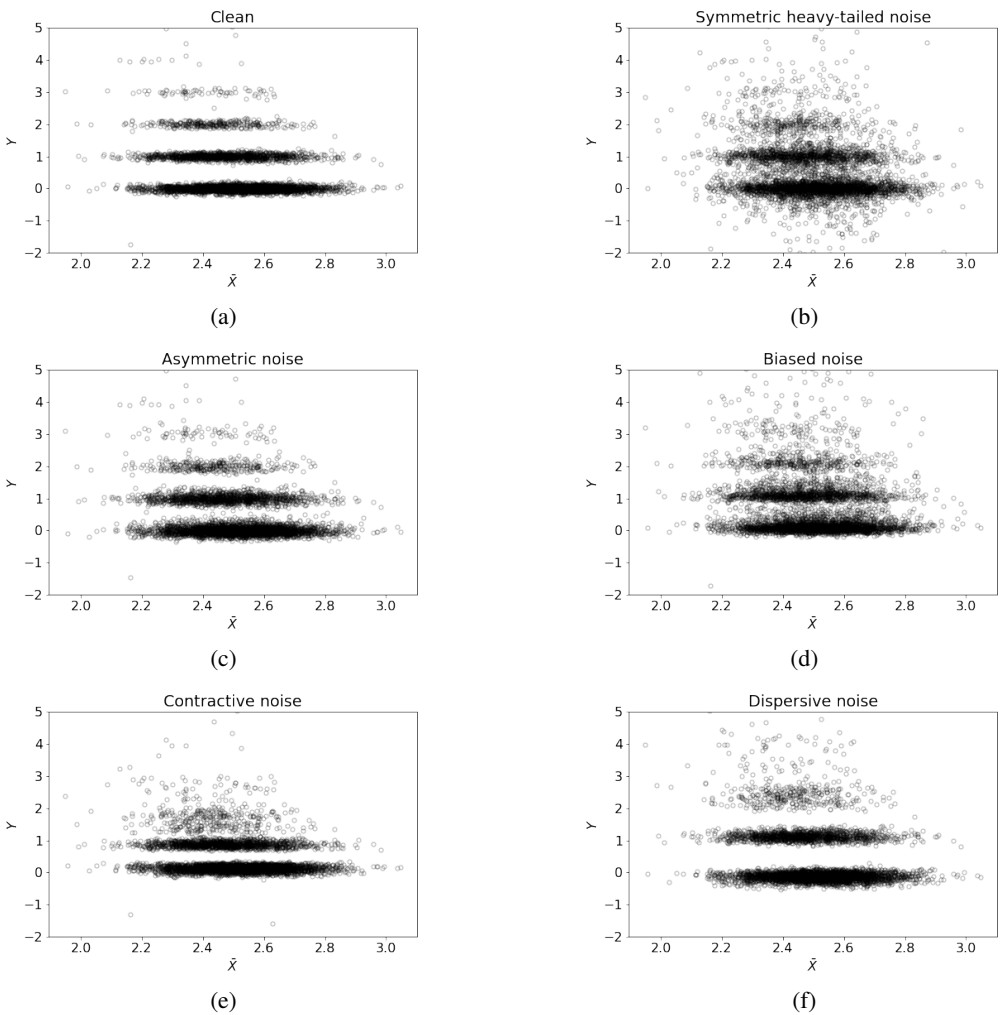

Figure A2: **Illustration of the generated data with different corruptions.** (a): Clean samples. (b): Samples with symmetric heavy-tailed noise. (c): Samples with asymmetric noise. (d): Samples with biased noise. Noise magnitude is set to 0.1. (e): Samples with contractive noise. (f): Samples with dispersive noise.

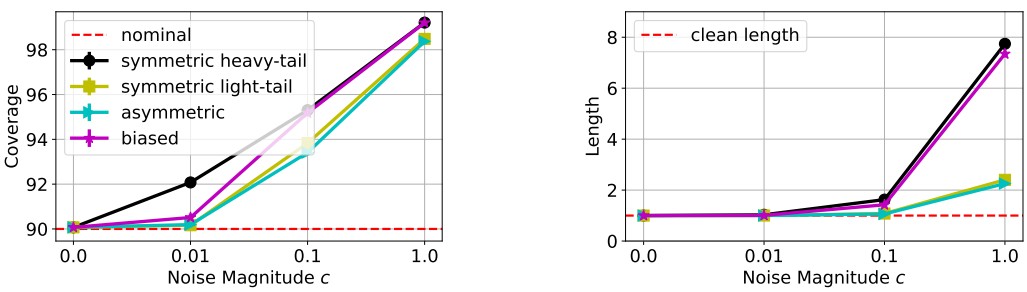

Figure A3: **Response-independent noise.** Performance of conformal prediction intervals with target coverage $1 - \alpha = 90\%$, using a clean training set and a noisy calibration set. Left: Marginal coverage; Right: Length of predicted intervals (divided by the average clean length) using symmetric, asymmetric and biased noise with a varying magnitude. Other details are as in Figure 3.

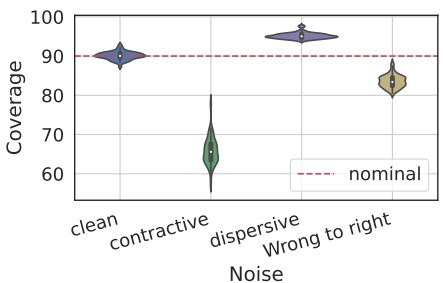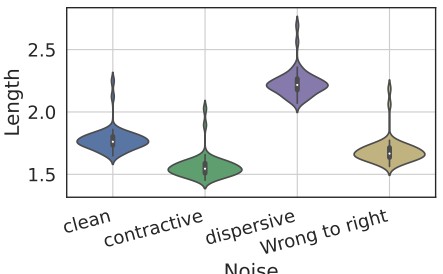

Figure A4: **Response-dependent noise.** Performance of conformal prediction intervals with target coverage $1 - \alpha = 90\%$, using a clean training set and a noisy calibration set. Left: Marginal coverage; Right: Length of predicted intervals. The results are evaluated over 50 independent experiments.

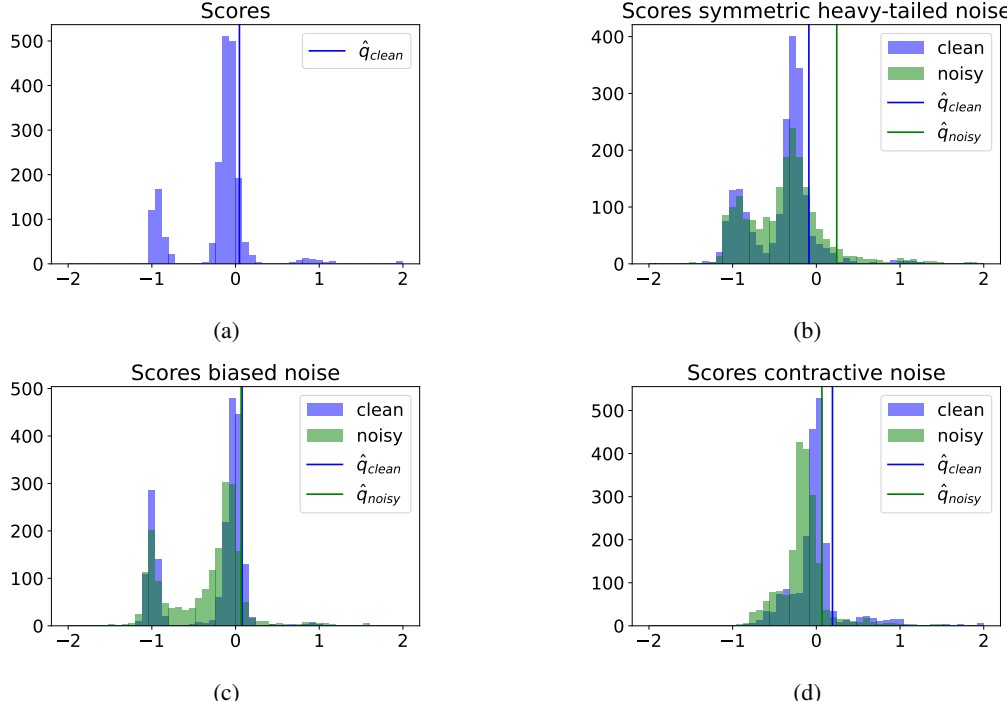

Figure A5: **Illustration of the CQR scores**. (a): Clean training and calibration sets. (b): Symmetric heavy-tailed noise. (c): Biased noise. Noise magnitude is set to 0.1. (d): Contractive noise. Other details are as in Figure 3.

### A2.3 REGRESSION: AESTHETIC VISUAL RATING

Herein, we provide additional details regarding the training of the predictive models for the real world regression task. As explained in Section 4.1, we trained two different models in this experiment. The first is a quantile regression model for CQR. Here we trained the model for 70 epochs using 'SGD' optimizer with a batch size of 128 and an initial learning rate of 0.001 decayed every 20 epochs exponentially with a rate of 0.95 and a frequency of 10. We applied dropout regularization to avoid overfitting with a rate of 0.2. The second model is a classic regression model for conformal with residual magnitude scores. Here we trained the model for 70 epochs using 'Adam' optimizer with a batch size of 128 and an initial learning rate of 0.00005 decayed every 10 epochs exponentially with a rate of 0.95 and a frequency of 10. The dropout rate in this case is 0.5.

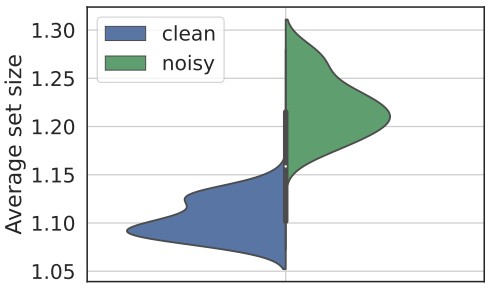

Figure A6: **Effect of label noise on CIFAR-10**. Distribution of average prediction set sizes over 30 independent experiments evaluated on CIFAR-10H test data using noisy and clean labels for calibration. Other details are as in Figure 1

### A2.4    CLASSIFICATION: OBJECT RECOGNITION EXPERIMENT

Here we provide additional results of the classification experiment with CIFAR-10H explained in Section 4.2. We apply conformal prediction with the APS score. The marginal coverage achieved when using noisy and clean calibration sets are depicted in Figure 1. Figure A6 illustrates the average prediction set sizes that are larger when using noisy data for calibration and thus lead to higher coverage level.

