# OpenReview forum: "Conformal Prediction is Robust to Label Noise"
_ICLR.cc/2023/Conference — Submitted to ICLR 2023_

### Official Review · Reviewer_mtdD · 2022-10-25

**Confidence:** 3
**Correctness:** 3
**Technical Novelty And Significance:** 2
**Empirical Novelty And Significance:** 3
**Recommendation:** 6

**Clarity, Quality, Novelty And Reproducibility:**

The proofs are densely written and quite hard to follow.
See some details in the section below.

**Strength And Weaknesses:**

The topic of the paper is highly interesting and relevant to the AI/ML community.
My main concern is that the text as written is quite confusing and I am not sure I understand the significance of their results. The assumptions seem to be very strong and unverifiable, contrary to some of the (over) claims in the text. See more details below.

**Summary Of The Paper:**

The paper focuses on the situation where the training data are corrupted by noise, which often happens in practice, and evaluates to what extent these nuisances affect the uncertainty quantification based on the conformal prediction. The main conclusion is that one should not worry in practical cases.

**Summary Of The Review:**

1) The reference to the probability distribution at play is quite confusing.
   - In Equation 1, P refers to the tensorized probability over the calibration and test data.
   - The same P is used in equation 2, however C_noisy involves the corrupted label that does not follow the same distribution
   - The same P also defined the oracle classifier P(tilde Y | X) where no other calibration data are involved.

2) Why is the assumption that access to the oracle classifier is reasonable in any way? It is not testable (oracle is never available) and this seems to be equivalent to the trained model classifying perfectly any noisy data. In that case, it seems quite natural that building a conformal prediction set based on a perfect classifier will behave perfectly. I am not sure to understand the insights/meaning of this result.

3) Taking into account variation in the labels due to noise, seems to be a special case of distribution shift. How does the current contribution differ from the conformal prediction that accounts for such a shift? Also, the label noise seems to not affect a conformal set whose underlying model is invariant to slight variation in the labels. Does this relate to a conformal set based on stability assumptions?

4) Some points on the Proofs.
In the proof of Lemma A1, the authors might want to clarify a bit formally “hat f be a classifier that ranks the classes in the same order as the oracle model” and also how is this precisely used in the proof. Seems to be in the monotonicity assumption but it is unclear and hard to follow.
Similarly, the proof of proposition 5 is very unclear. Y and tilde Y are discrete random variables, assuming a density for their distribution is a bit too strong here. Also, the Proposition talks about any tilde Y not equal to Y (in distribution) but the proof assumes a substantially strong hypothesis. For example, the set A can be empty depending on p and tilde p. The design of the adversarial score is also unclear: the set A is independent of the input data x (but this is ok) and also independent of any output y. Then the sequence of s(x_i, y_i) are not exchangeable (and so not valid scores) since the score is invariant wrt to the data. The rest of the proof is also very confusing.

---

> ### Author Response · Authors · 2022-11-17
> **Response to Reviewer mtdD**
>
> We thank the reviewer for all the comments and for putting the time and effort into improving our manuscript. We carefully responded to the reviewer's criticisms point-by-point, in a response document that we have attached to the "Supplementary Material'' section of our revised submission.
>
> $\textbf{Please download the .zip file from the Supplementary Material and open the ``response.pdf'' file to view our response.
>  }$
>
> We uploaded a revised version of our manuscript, replacing the original submission, and at the end of the "response.pdf'' file we included a marked-up version of our submission to track the changes easily.

---

> > ### Author Response · Authors · 2022-12-04
> > **Response to Reviewer mtdD**
> >
> > Dear Reviewer,
> >
> > Thank you for your efforts in reviewing our paper. We kindly remind you that the discussion period ends in eight days. We would sincerely appreciate any follow-up on our response. This way, we will know that you have seen it, and we will be able to address any further concerns or questions that you may have.
> >
> > Thank you,
> >
> > The Authors

---

> > > ### Comment · Reviewer_mtdD · 2022-12-08
> > > **Comment on rebuttal**
> > >
> > > Thanks for the rebuttal and update of the paper. It clarifies some of my misunderstanding.
> > > However, I still think that the interest of the results of this paper is quite limited because we can't test the hypothesis to make sure that the CP noisy is well calibrated. By definition, we don't have access to the oracle and therefore the hypotheses of the theorems stated don't seem to be verifiable in practice.
> > >
> > > That said, the paper sheds light on particular conditions, admittedly theoretical, for which one can ignore the ambient noise in the labels when designing conformal predictions. This does not seem recommendable to me, and even dangerous, especially since the conditions of application are largely unclear and not verifiable.
> > >
> > > Still, I raised my score. Theoretical works that advance the understanding of properties of CP are interesting in my opinion.

---

> > > > ### Author Response · Authors · 2022-12-11
> > > > **Response to Reviewer mtdD**
> > > >
> > > > Thanks for improving your score! Hopefully our paper will help people realize when they can and cannot use conformal.
> > > >
> > > > In real life, labels are almost always noisy, so practitioners will need to think carefully about whether conformal prediction will work --- even though the setting involves unverifiable assumptions!
> > > > We hope that our paper gets the message across that a) if the model is good AND b) the noise is dispersive, then conformal can be run; and conversely, if either condition is violated (as in the `mean-reversion’ setting), conformal cannot be run.
> > > >
> > > > The reviewer should let us know if they see a way for our paper to get the message across more clearly. it is worth noting that the conditions a) and b) were unknown before our work, but they do seem to be intrinsic to conformal.

---

### Official Review · Reviewer_tBnf · 2022-10-26

**Confidence:** 4
**Clarity, Quality, Novelty And Reproducibility:** The paper is well written, with both …
**Correctness:** 4
**Technical Novelty And Significance:** 3
**Empirical Novelty And Significance:** 4
**Recommendation:** 6

**Strength And Weaknesses:**

1.	I have some concerns with the assumption of Section 2.1. The labels of images are not usually randomly corrupted. For example, dogs are often misclassified as cats, and vice versa. The confusion matrix case looks more reasonable. However, it is not supported by any theoretical results.

2. The second formula in Section 2.1 involves \hat{\pi}_y(x), which is never mentioned before use. Similarly, I can guess the meaning of \hat{C}_{\rm noisy}^{\rm APS} in proposition (1). The authors should define the notations before use.

3. In Figures 1, 2, 4, and 5, there are some bars between distributions. The authors didn’t explain what they mean. The authors only mentioned the error bar in the caption of Figure 3.

4. Possible typos:
1) The second formula of section 2.1: \hat{\pi}_y(x)_y. The last subscript y looks redundant.
2) The last sentence of the second last paragraph of section 3.1: "be more robust to label noise then HPS". Should it be "than"?


**Summary Of The Paper:**

The paper addressed an interesting question: is conformal prediction (CP) robust to label noise? They explored several situations including both regression and classification cases. They also constructed the corresponding theories to support their claim. The authors also included experiments on both synthetic and real examples, demonstrating the robustness of conformal prediction. The paper is easy to follow, and the writing is fluent.

**Summary Of The Review:**

The topic that analyzing whether conformal prediction is trustful when label noise exists is important. Oveall, the authors provide solid study to address this question.

---

> ### Author Response · Authors · 2022-11-17
> **Response to Reviewer tBnf**
>
> We thank the reviewer for all the comments and for putting the time and effort into improving our manuscript. We carefully responded to the reviewer's criticisms point-by-point, in a response document that we have attached to the "Supplementary Material'' section of our revised submission.
>
> $\textbf{Please download the .zip file from the Supplementary Material and open the ``response.pdf'' file to view our response.
>  }$
>
> We uploaded a revised version of our manuscript, replacing the original submission, and at the end of the "response.pdf'' file we included a marked-up version of our submission to track the changes easily.

---

> > ### Author Response · Authors · 2022-12-04
> > **Response to Reviewer tBnf**
> >
> > Dear Reviewer,
> >
> > Thank you for your efforts in reviewing our paper. We kindly remind you that the discussion period ends in eight days. We would sincerely appreciate any follow-up on our response. This way, we will know that you have seen it, and we will be able to address any further concerns or questions that you may have.
> >
> > Thank you,
> >
> > The Authors

---

### Official Review · Reviewer_DjSE · 2022-10-27

**Confidence:** 2
**Correctness:** 4
**Technical Novelty And Significance:** 3
**Empirical Novelty And Significance:** 2
**Recommendation:** 5

**Clarity, Quality, Novelty And Reproducibility:**

The paper is well organized, and the idea of studying the robustness of conformal prediction to label noise is relatively novel.

**Strength And Weaknesses:**

Strength: This paper studies a very important problem and tries to provide some theoretical results about the label noise learning tasks.

Weakness: The paper does not provide much explanation for how the findings in this paper can be applied to existing noise-label learning methods.
The adopted datasets are relatively small, there are real-world noise datasets such as Clothing1M and Webvision that are widely used for noisy label learning but are not adopted in this paper.

**Summary Of The Paper:**

This paper studies the robustness of conformal prediction to label noise, characterizing when and how it is possible to construct uncertainty sets that correctly cover the unobserved noiseless ground truth labels.

**Summary Of The Review:**

The idea of studying the robustness of conformal prediction to label noise is relatively novel. However, The paper does not provide much explanation for how the findings in this paper can be applied to existing noise-label learning methods. Moreover, the adopted datasets are relatively small.

---

> ### Author Response · Authors · 2022-11-17
> **Response to Reviewer DjSE**
>
> We thank the reviewer for all the comments and for putting the time and effort into improving our manuscript. We carefully responded to the reviewer's criticisms point-by-point, in a response document that we have attached to the "Supplementary Material'' section of our revised submission.
>
> $\textbf{Please download the .zip file from the Supplementary Material and open the ``response.pdf'' file to view our response.
>  }$
>
> We uploaded a revised version of our manuscript, replacing the original submission, and at the end of the "response.pdf'' file we included a marked-up version of our submission to track the changes easily.

---

> > ### Author Response · Authors · 2022-12-04
> > **Response to Reviewer DjSE**
> >
> > Dear Reviewer,
> >
> > Thank you for your efforts in reviewing our paper. We kindly remind you that the discussion period ends in eight days. We would sincerely appreciate any follow-up on our response. This way, we will know that you have seen it, and we will be able to address any further concerns or questions that you may have.
> >
> > Thank you,
> >
> > The Authors

---

### Official Review · Reviewer_snZB · 2022-11-02

**Confidence:** 4
**Correctness:** 3
**Technical Novelty And Significance:** 2
**Empirical Novelty And Significance:** 2
**Recommendation:** 3

**Clarity, Quality, Novelty And Reproducibility:**

The paper is clearly written, but the contribution may not be solid enough. See comments above.

**Strength And Weaknesses:**

Strength. The writing is quite clear. The topic is interesting as corrupted labels are commonly seen in real data sets. The discussion of whether some standard techniques based on well-specified models are applicable in general to real noisy data is always appealing.

Weakness. However, this paper is insufficient in several aspects:
1. The biggest concern of mine is that this paper only presents in all positive cases, the coverage set is valid (>= 1-alpha). However, one of the most important aspects of conformal prediction is to show the tightness of the coverage. One can always return a valid coverage set by applying a trivial one. One missing part is the analysis of how much the noise inflated the coverage set.

2. The second concern of mine is that the theoretical results are built upon a very simple noise mechanism. For instance, the unimodal assumption is kind of strong. Also, the assumption on Z is not very commonly seen, more explanation is appreciated.

3. The empirical result on real data is too simple. More datasets in large scale should be discussed.

4. One minor point, in practice, domain shift is also commonly seen. It would be interesting to see some discussion on that.

**Summary Of The Paper:**

This paper studies whether a coverage set is still valid under label noise. They study both regression and classification cases. Empirically, they demonstrated in some experiments that constructing coverage sets based on examples with corrupted labels is still valid as long as the noise is benign (not too adversarial). Theoretically, several noisy labeled cases are studied, for example, in 2.1, a commonly used corruption model is used to show their point, and in 2.2, a shifted noise is added for the regression model, where the noise is symmetric unimodal, e.g. standard normal. They also show in 2.4 that when noise is not benign, the coverage sets built upon noisy examples may not be valid anymore. Lastly, synthetic and real experiments are shown.

**Summary Of The Review:**

See comments above.

---

> ### Author Response · Authors · 2022-11-17
> **Response to Reviewer snZB**
>
> We thank the reviewer for all the comments and for putting the time and effort into improving our manuscript. We carefully responded to the reviewer's criticisms point-by-point, in a response document that we have attached to the "Supplementary Material'' section of our revised submission.
>
> $\textbf{Please download the .zip file from the Supplementary Material and open the ``response.pdf'' file to view our response.
> }$
>
> We uploaded a revised version of our manuscript, replacing the original submission, and at the end of the "response.pdf'' file we included a marked-up version of our submission to track the changes easily.

---

> > ### Author Response · Authors · 2022-12-04
> > **Response to Reviewer snZB**
> >
> > Dear Reviewer,
> >
> > Thank you for your efforts in reviewing our paper. We kindly remind you that the discussion period ends in eight days. We would sincerely appreciate any follow-up on our response. This way, we will know that you have seen it, and we will be able to address any further concerns or questions that you may have.
> >
> > Thank you,
> >
> > The Authors

---

### Decision · Program_Chairs · 2023-01-20

**Decision:**

Reject

**Justification For Why Not Higher Score:**

The manuscript needs a lot of changes that need a second round of review.

**Justification For Why Not Lower Score:**

N/A

**Metareview: Summary, Strengths And Weaknesses:**

The paper study the robustness of conformal prediction. Their analysis tackles regression and classification problems, characterizing when and how it is possible to construct uncertainty sets that correctly cover the unobserved noiseless ground truth labels. They argue that naïve conformal prediction covers the noiseless ground truth label unless the noise distribution is adversarially designed. They correct for the noise of bounded size in the conformal prediction algorithm to ensure correct coverage of the ground truth labels without a score or data regularity.

++ The paper writing is generally clear. The problem can be interested in the community.

-- The conformal prediction in the paper seems not to show the tightness of the coverage, while the theoretical results are built on a simple noise mechanism.

-- The empirical result on real data is simple. Need more datasets on a large scale.

-- The confusion matrix noise model is missing in the paper, although the authors discussed it in the response pdf. But it looks incomplete.

-- The proof of the results in the paper needs a lot of improvement. It seems it needs to be modified a lot compared to the original submission.

After carefully reading the paper, the response pdf, and the reviews, the meta-reviewer cannot recommend acceptance at the stage. This manuscript needs lots of improvement before publishing.